# Analysis of Agricultural Water Use Efficiency Based on Analytic Hierarchy Process and Fuzzy Comprehensive Evaluation in Xinjiang, China

**Peibing Song [1], Xiaoying Wang [2], Chao Wang [3,\*], Mengtian Lu [1], Lei Chen [4], Lingzhong Kong [5], Xiaohui Lei [3] and Hao Wang [1,3]**

1. College of Civil Engineering and Architecture, Zhejiang University, Hangzhou 310058, China; songpeibing@zju.edu.cn (P.S.); 11512052@zju.edu.cn (M.L.); wanghao@iwhr.com (H.W.)
2. College of Water Resource Science and Engineering, Taiyuan University of Technology, Taiyuan 030024, China; wxy769662707@163.com
3. State Key Laboratory of Simulation and Regulation of Water Cycle in River Basin, China Institute of Water Resources and Hydropower Research, Beijing 100038, China; lxh@iwhr.com
4. School of Water Resources and Environment, China University of Geosciences (Beijing), Beijing 100083, China; chenleit01@163.com
5. College of Hydraulic Science and Engineering, Yangzhou University, Yangzhou 225009, China; lzkong@126.com
* Correspondence: wangchao@iwhr.com; Tel.: +86-10-6878-5503

**Abstract:** Improving agricultural water use efficiency (WUE) and reducing the proportion of agricultural water consumption are of great significance for coping with the water crisis in the world, particularly in northwest China. However, considering the lack of WUE indicators for the selection and an evaluation of system construction in Xinjiang, the implementation of the strictest water resources management system in this province has been seriously restricted. In order to evaluate the agricultural water utilization situation of 14 administrative regions in Xinjiang, a synthetical evaluation model is proposed combining the analytic hierarchy process method and the fuzzy evaluation method. Firstly, an evaluation system composed of the target layer (i.e., water use efficiency), the criterion layer (i.e., water use situation, engineering measure, planting structure) and the index layer (e.g., utilization coefficient of irrigation water, weighting irrigation quota, grain production per cubic meter of irrigation water, etc.) is constructed. Secondly, a classification standard of evaluation indicators is put forward and a fuzzy evaluation model is introduced into calculate agricultural WUE. Finally, key evaluation indicators that lead to these administrative regions with very low WUE are identified, and discussions on improving agricultural WUE in Kashgar are made. The evaluation results of this study are beneficial for providing support for reforming agricultural water use and promoting sustainable agricultural development in Xinjiang.

**Keywords:** agricultural water management; water use efficiency (WUE); spatial feature; expert scoring; Xinjiang

## 1. Introduction

Water is the source of life and the important substantial foundation of human survival and social development [1,2]. It is also an irreplaceable natural resource for agricultural production, especially in arid regions in China [3,4]. In the past decade, agricultural water consumption accounted for approximately 70–75% of global water consumption, while China's agricultural water consumption accounted for 61.2–63.6% of national water consumption [5,6]. Other data showed that agricultural

production accounted for approximately 92% of the global annual average water footprint between 1996 and 2005, by means of a direct way or an indirect way [7]. In China, the agricultural water stress index, the established water footprint framework and blue-green water resources, increased from 0.32 (mid-water stress level) in 2000 to 0.49 (high water stress level) in 2015 [8]. According to China Statistical Yearbook, the irrigated area of cultivated land increased from 59.3 million ha in 2009 to 68.3 million ha in 2018, with an increase of 15.2% over the last ten years. With the rapid development of industrialization and urbanization, China has been confronted with a serious crisis of water resources, which have caused more effects on the biosphere at a striking speed, especially on crops [9,10]. Moreover, uncertainties involving impacts of climate change and natural hazards on crop yields will be large, and uneven spatial patterns of precipitation changes may enhance drought impacts in the 21st century [11,12]. Studies demonstrated that agricultural droughts may become more severe for 2020–2049, relative to 1971–2000, and the frequency of long-term agricultural droughts would become more frequent compared to those of short-term droughts [13]. Based on IPCC-AR5 scenarios, agricultural water consumption may increase to about 4% of total water consumption along with rising temperatures of 1 °C in North China, which may bring great challenges to food security, particularly for poorer populations [14,15]. As a consequence, it is noticeable to increase the risk of water shortage for agriculture as evapotranspiration increases dramatically in the context of global warming [16,17]. As an important agricultural production base in China, the water resource shortage in Xinjiang has become a critical factor that makes the conflict between local socio-economic development and ecological balance more acute, and agricultural water consumption has occupied more than 95% of total socio-economic water consumption in the past decades [18,19]. In order to implement the strictest water resources management system and realize sustainable agricultural development in Xinjiang, it is essential to increase agricultural water use efficiency (WUE) and tap water-saving potential, given the limited available water resources [20].

In general, evaluation of WUE is a multi-objective and multi-criterion synthetical problem in essence [21]. In terms of screening evaluation indicators and constructing evaluation models, scholars at home and abroad have carried out massive application studies [22]. Since 1979, the International Commission on Irrigation and Drainage (ICID), the American Society of Civil Engineers (ASCE), the Food and Agriculture Organization of the United Nations (FAO) and the Irrigation Association of Australia (IAA) have put forward the definition of irrigation efficiency successively [23–27]. Other evaluation indicators, such as the canal WUE and the field WUE, can also be used to represent agricultural WUE [28,29]. Cao et al. established the generalized efficiency (GE) to assess the effective use rate of generalized water resources, and this index was defined as the ratio of total water consumption to total water resources [20]. Sun et al. mitigated the impact of industrial structure and labor division on WUE, by empowering the original indexes and proposing the synthetic evaluation indexes [30]. A water debt repayment time indicator was introduced by Tuninetti et al., measuring the time required to replenish water resources used for annual crop production [31]. In addition, the farmland total WUE, the crop water productivity and the real irrigation WUE have been extensively adopted to describe agricultural WUE [32,33]. Natural factors and human factors are significant factors affecting agricultural WUE. The former factors are related to natural conditions, including climate, temperature and soil, as well as farmland infrastructure construction, while the latter factors can be divided into engineering factors, management factors and technical factors [34–36]. Kang et al. analyzed the evolution of irrigation water productivity and its relationships with variations of crop yield, cropping patterns and fertilization [37]. Wang et al. revealed agricultural investment, economic growth, industrial restructuring and agricultural plant structural adjustment had remarkable effects on agricultural WUE, by means of the Tobit model [38]. Lin et al. researched that plastic film mulch was widely applied into spring maize for its improving WUE and increasing crop yield, compared to no mulching [39].

When it comes to the evaluation model, data envelopment analysis (DEA), stochastic frontier analysis (SFA), slack based measure (SBM), projection pursuit (PP), analytic hierarchy process (AHP) and driver-pressure-state-impact-response (DPSIR) models have been widely used for constructing

evaluation systems. The information significance difference (ISD) model combined with the DPSIR model, was applied by Liu et al. in the Sanjiang Plain of northeastern China, and this method was able to reduce the number of evaluation indicators from 44 to 14 [40]. The data envelopment analysis method was employed by Geng et al. to quantify agricultural WUE under two scenarios, in which the CCR model and BCC model with different assumptions attracted great attention [5]. Based on the panel data from 2000 to 2015, the main factors influencing agricultural WUE were identified by using the stochastic frontier analysis model and spatial econometric model [35]. In order to reveal the great variety of regional water use, a projection pursuit cluster model and the accelerating genetic algorithm were applied to calculating the overall efficiency of 31 provinces in China [41]. A common analytic hierarchy process method was modified by taking hierarchical structure and the number of indices into consideration, so as to improve the accuracy and reliability of the calculated weight [42]. With the development of satellite-based remote sensing techniques, gross primary production and evapotranspiration provided by MODIS products were used to analyze spatial and temporal patterns of WUE, as well as its relationships with environmental factors [33,43]. Compared to other evaluation methods, the fuzzy evaluation method has the advantage of clear results and a strong system, and it is mostly adopted to solve these problems that are fuzzy and difficult to quantify, especially non-deterministic problems [44]. Moreover, fuzzy mathematics is capable of taking fuzzy objects to be examined and fuzzy concepts reflecting fuzzy objects as certain fuzzy sets, and establishing appropriate membership functions [45]. Apart from a single evaluation method, some coupled methods were proposed in assessment of agricultural WUE, because these methods are likely to take advantage of multiple evaluation methods [46]. Taking the fuzzy analytic hierarchy process (FAHP) method as an example, the analytic hierarchy process is flexible to applicate and suitable for handling complex real-life problems, while fuzzy decision theory can make up for deficiencies and limitations, including uncertainty and poor reliability of determining index weights [44]. In spite of the good performance of the fuzzy analytic hierarchy process method, there is still room to deal with the nonlinear problem [47]. Accordingly, a coupled method composed of fuzzy comprehensive evaluation and the analytic hierarchy process has been acknowledged to be effective towards investigating agricultural WUE.

Aimed at quantifying agricultural water use security of 14 administrative regions in Xinjiang Uygur Autonomous Region (Xinjiang), this study is organized as follows: (1) screening evaluation indicators and establishing an evaluation system based on the analytic hierarchy process method; (2) putting forward a classification standard of evaluation indicators and introducing a fuzzy evaluation model into calculating WUE; (3) identifying key evaluation indicators that lead to very low WUE in some administrative regions, and taking Kashgar as a case study to discuss how to improve agricultural WUE.

## 2. Materials and Methods

### 2.1. Study Area

As the northwest border area of China and the center of Eurasia, Xinjiang is located within 73°40′−96°18′ E and 34°25′−48°10′ N, and it covers a land area of over 1.66 million km$^2$ [48]. The north of Xinjiang is the Altai Mountains, and the south of that is the Kunlun Mountains. The Tianshan Mountains lie in the central part of Xinjiang, while the Tarim Basin and the Turpan-Hami Basin are located in the south and southeast of Xinjiang, respectively. Due to unique topography, Xinjiang is divided into three regions: Southern Xinjiang, Northern Xinjiang and Eastern Xinjiang, with land areas of 1.05 million km$^2$, 0.4 million km$^2$ and 0.21 million km$^2$, respectively [49]. At present, Xinjiang administers 4 prefecture-level cities (i.e., Urumqi, Karamay, Turpan, Hami), 5 prefectures (i.e., Aksu, Kashgar, Hotan, Tacheng, Altay), 5 autonomous prefectures (i.e., Changji, Bortala, Bayingol, Yili, Kizilsu), and 10 county-level cities under the jurisdiction of the autonomous region directly. The location of the study area is shown in Figure 1.

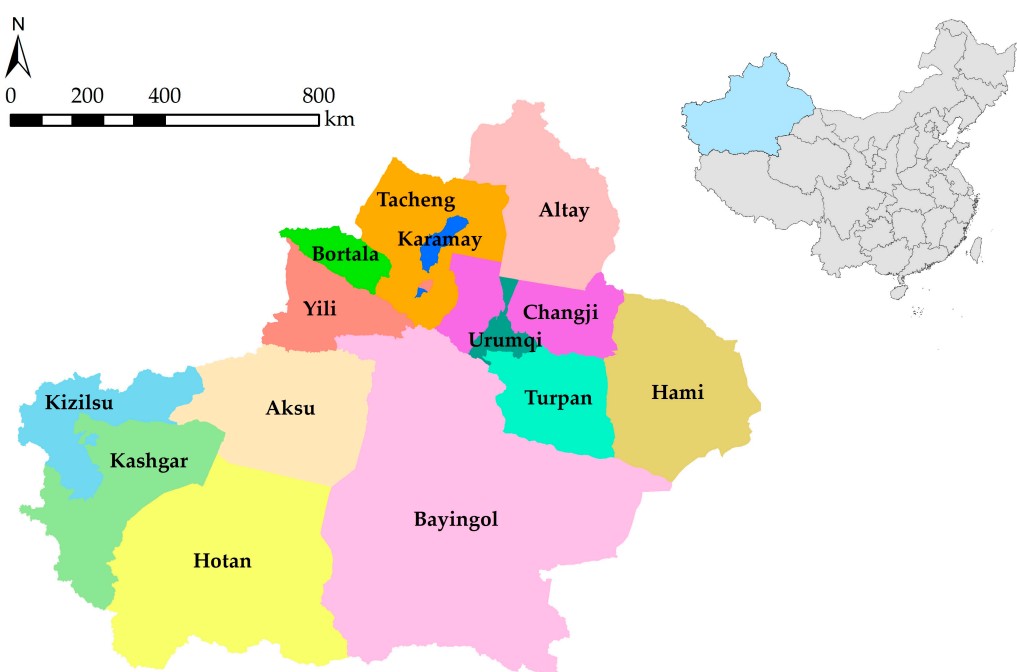

**Figure 1.** The location of the study area.

There are more than 570 rivers in Xinjiang, 97% of which originate from mountains. Annual runoff of surface water in Xinjiang is 88.4 billion m$^3$, accounting for about 3% of the national total. Per capita water resources in Xinjiang are 5500 m$^3$, and are 2.25 times the national average. The annual precipitation of Xinjiang is 254.4 billion m$^3$ (1956–2000 series), equivalent to the precipitation depth of 154 mm. Regional distribution of annual precipitation tends to be more in northern Xinjiang than in southern Xinjiang, more in the west than in the east, more mountains than plains and more on windward slopes than leeward slopes. Annual precipitation in the mountains, basin edges and basin centers in northern Xinjiang is generally 400–800 mm, 150–200 mm and 100 mm, respectively, while that in southern Xinjiang is 200–500 mm, 50–80 mm and 20 mm, respectively. Distribution tendency of evaporation capacity are small in the north and large in the south, small in the west and large in the east and small in the mountains and large in the plains. Evaporation capacity in mountain areas is 800–1200 mm, while that in plain basins is 1600–2200 mm. With a typical temperate continental arid climate, Xinjiang has an annual mean temperature ranging from 6.2 to 9.0 °C at the rate of 0.32 °C/decade, and an annual rainfall varying from 93.2 to 205.8 mm at the rate of 8.23 mm/decade during 1960–2013 [50]. Mean annual temperature over 1991–2000 was higher than that over 1961–1990, with an increase of 0.8 °C in the north and 0.5 °C in the south [51]. In China, annual insolation duration in Xinjiang is second only to that in Tibetan Plateau, reaching 2600–3400 h per year.

## *2.2. Methods*

### 2.2.1. Constructing an Evaluation Indicator System

In order to fully reflect the situation of agricultural WUE in Xinjiang, eight evaluation indicators are screened from Water Resources Bulletin of Xinjiang Uygur Autonomous Region, Xinjiang Statistical Yearbook, General plan of efficient water-saving irrigation in the 13th five year plan and Agricultural irrigation water quota of Xinjiang Uygur Autonomous Region. These evaluation indicators can be expressed as follows:

C1 (the utilization coefficient of irrigation water): The ratio of the amount of irrigation water available for crops to the total amount of irrigation water.

C2 (the weighting irrigation quota, m$^3$/mu): Comprehensive irrigation water consumption per unit irrigation area.

C3 (the grain production per cubic meter of irrigation water, kg/m$^3$): An indicator adopted to measure agricultural water resources productivity.

C4 (the proportion of agricultural water, %): The ratio of agricultural irrigation water consumption to total water consumption of national economy and society.

C5 (the agricultural water consumption for CNY ten thousand output value, m$^3$): The ratio of agricultural irrigation water consumption to total agricultural output value.

C6 (the water-saving irrigation rate): The ratio of water-saving irrigation project area to total irrigation area.

C7 (the planting proportion of crops, %): The ratio of the planting area of grain crops to total planting area of crops.

C8 (the water-consumption proportion of crops): The ratio of the water consumption of grain crops to agricultural water consumption.

Based on indicators mentioned above, a hierarchical structure for evaluating agricultural WUE is proposed, including target layer, criterion layer and index layer. The target layer refers to agricultural WUE. The criterion layer is composed of water use situation (B1), engineering measure (B2) and planting structure (B3). The evaluation of water use situation (B1) is able to synthetically represent the level of agricultural production in the irrigation area, and it contains five indicators, namely, C1, C2, C3, C4 and C5. The evaluation of engineering measure (B2) provides the basis for improving agricultural WUE, and it contains one indicator: C6. The evaluation of planting structure (B3) can be used to reduce non-engineering water loss and it contains two indicators: C7 and C8. It is also worth pointing out that this hierarchical structure follows two principles: one is that there should be no overlap between any two indicators, and the other is that indicators listed in the lower layer should include content of corresponding upper layer as much as possible. The evaluation system constructed by analytic hierarchy process (AHP) is shown in Figure 2.

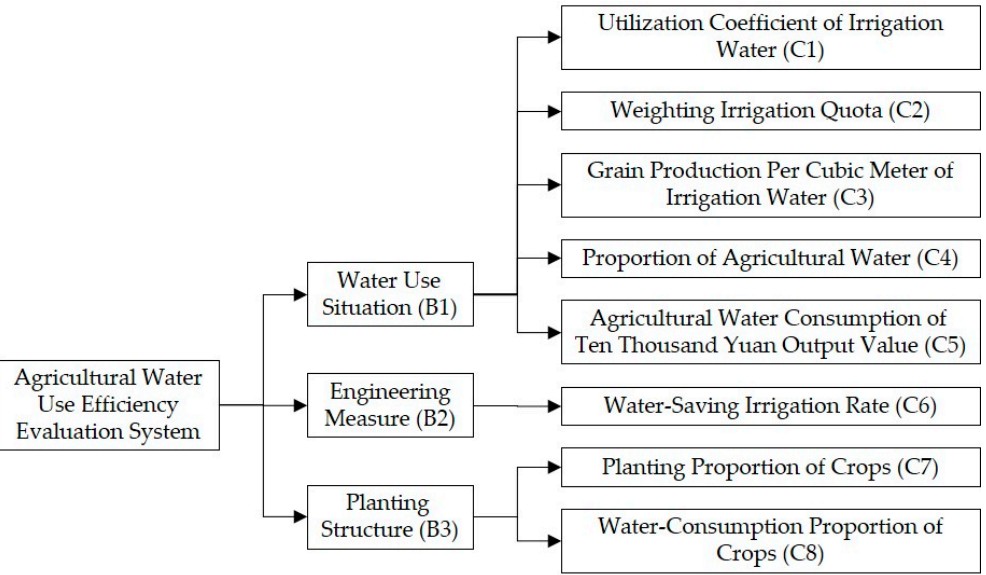

**Figure 2.** Evaluation system of agricultural water use efficiency in Xinjiang.

Taking the indicator C1 (the utilization coefficient of irrigation water) as an example, its calculation equation can be expressed as follows:

$$\eta_R = \frac{\eta_L \times V_L + \eta_M \times V_M + \eta_S \times V_S + \eta_W \times V_W}{V_L + V_M + V_S + V_W} \tag{1}$$

where $\eta_R$ denotes the utilization coefficient of irrigation water. $\eta_L$, $\eta_M$, $\eta_S$ and $\eta_W$ denote the mean utilization efficiency of irrigation water in large, medium, small and well irrigation areas, respectively.

$V_L$, $V_M$, $V_S$ and $V_W$ denote the gross irrigation water in large, medium, small and well irrigation areas, respectively, $10^4$ $m^3$. Parameters $\eta$ and $V$ mentioned above can be obtained from Calculation and analysis report on the effective utilization coefficient of irrigation water in Xinjiang in 2016.

For Urumqi, $\eta_{Urumqi} = 2415.26/4566.5 = 0.529$. In the same say, the indicator C1 in other regions can be obtained. With reference to the definition of other evaluation indicators, all indicators values of 14 administrative regions (i.e., four prefecture-level cities, five prefectures, five autonomous prefectures) can be calculated similarly, as shown in Table 1.

**Table 1.** Evaluation indicators values of 14 administrative regions.

| Regions | C1 | C2 ($m^3$/mu) | C3 (kg/$m^3$) | C4 (%) | C5 ($m^3$) | C6 | C7 (%) | C8 |
|---------|------|------|------|------|------|-------|------|-------|
| Urumqi | 0.529 | 568 | 0.90 | 57.3 | 2374 | 0.174 | 32.8 | 0.575 |
| Karamay | 0.530 | 490 | 1.26 | 59.6 | 8416 | 0.741 | 13.2 | 0.561 |
| Turpan | 0.605 | 672 | 0.49 | 92.7 | 2891 | 0.478 | 7.5 | 0.383 |
| Hami | 0.632 | 436 | 0.82 | 85.9 | 2695 | 0.682 | 35.6 | 0.410 |
| Changji | 0.585 | 380 | 1.30 | 92.4 | 1836 | 0.680 | 52.4 | 0.547 |
| Bortala | 0.589 | 486 | 1.57 | 95.8 | 2550 | 0.798 | 35.0 | 0.504 |
| Bayingol | 0.507 | 642 | 0.76 | 93.8 | 2856 | 0.645 | 16.1 | 0.464 |
| Aksu | 0.495 | 715 | 0.71 | 98.3 | 4947 | 0.266 | 26.3 | 0.467 |
| Kizilsu | 0.472 | 647 | 0.65 | 95.7 | 9422 | 0.122 | 73.0 | 0.231 |
| Kashgar | 0.466 | 755 | 0.57 | 97.7 | 5545 | 0.203 | 36.4 | 0.554 |
| Hotan | 0.459 | 849 | 0.50 | 96.7 | 7584 | 0.107 | 66.0 | 0.457 |
| Yili | 0.459 | 523 | 1.00 | 93.5 | 2740 | 0.251 | 76.7 | 0.679 |
| Tacheng | 0.459 | 365 | 1.82 | 97.0 | 2005 | 0.519 | 48.8 | 0.644 |
| Altay | 0.534 | 680 | 0.71 | 97.2 | 7469 | 0.784 | 35.9 | 0.459 |

### 2.2.2. Determining the Classification Standard of Evaluation Indicators

According to agricultural development direction in China, combined with the water utilization situation of Xinjiang, Technical standard for water-saving irrigation project and Agricultural irrigation water quota of Xinjiang Uygur Autonomous Region, the evaluation indicator value is divided into five grades: I, II, III, IV and V, corresponding to five classification standards of agricultural WUE: very low, low, medium, high and very high, respectively. Then, the grade of evaluation indicator values is listed in Table 2.

**Table 2.** The grade of evaluation indicator values (Pre denotes annual precipitation, mm).

| Evaluation Indicator | | I | II | III | IV | V |
|---|---|---|---|---|---|---|
| | C1 | ≤0.45 | 0.47 | 0.5 | 0.55 | ≥0.65 |
| C2 | Pre ≤ 400 | ≥875 | 750 | 625 | 500 | ≤375 |
| | 400 < Pre ≤ 800 | ≥625 | 500 | 375 | 250 | ≤125 |
| | C3 | ≤0.4 | 0.6 | 0.8 | 1.0 | ≥1.3 |
| | C4 | ≥97% | 95% | 93% | 90% | ≤80% |
| | C5 | 8000 | 6000 | 4000 | 2000 | ≤1000 |
| | C6 | ≤0.1 | 0.25 | 0.4 | 0.55 | ≥0.7 |
| | C7 | ≤0.1 | 0.25 | 0.4 | 0.55 | ≥0.7 |
| | C8 | ≥0.7 | 0.6 | 0.54 | 0.47 | ≤0.4 |

This grade can be regarded as significant signs reflecting the efficiency value of agricultural water utilization, with clear partition standards and discrimination boundaries. Specific descriptions of five classification standards are listed as follows:

(I) Very low WUE: the constructions of engineering facilities are few, the water-saving technologies are rarely used, the levels of water resource management are very low, and the water consumption indexes are obviously lower than the average level.

(II) Low WUE: the constructions of engineering facilities are less, the water-saving technologies are seldom used, the levels of water resources management are low and the water consumption indexes are lower than the average level.

(III) Medium WUE: the constructions of engineering facilities are at an average level, the water-saving technologies are adopted in the minority of areas, the levels of water resources management are medium and the water consumption indexes are at a medium level.

(IV) High WUE: the constructions of engineering facilities are in good condition, the water-saving technologies are adopted in the majority of areas, the levels of water resources management are high and the water consumption indexes are close to the advanced level.

(V) Very high WUE: the engineering facilities are complete and in good running direction, the water-saving technologies are adopted in the vast majority of areas, the levels of water resources management are very high and the water consumption indexes reach the advanced level.

The indicator C1 (the indicator value is represented by $C_{v,1}$) is selected as an example to explain its classification standard. If $C_{v,1} \leq 0.45$, it means that $C_{v,1}$ is at the grade I and this indicator has the characteristics of very low WUE; if $C_{v,1} \geq 0.65$, it means that $C_{v,1}$ is at the grade V and this indicator has the characteristics of very high WUE; if $0.45 < C_{v,1} < 0.47$, it means that $C_{v,1}$ is located between the grade I and II, and this indicator has the characteristics of very low and low WUE; and so on. As shown in Table 1, $C_{v,1}$ in Urumqi is 0.5289, between 0.5 and 0.55, which indicates that the C1 is located between the grade III and IV, and the utilization coefficient of irrigation water (C1) has the characteristics of medium and high WUE.

### 2.2.3. Fuzzy Comprehensive Evaluation Model

A fuzzy comprehensive evaluation is a scientific description of the fuzzy things, phenomena, and concepts in the transitional stage of the material system [52]. The specific steps of establishing a fuzzy evaluation model for calculating agricultural WUE are summarized as follows:

- Step 1. Construct a fuzzy membership matrix of evaluation indicators.

According to original values of indicators and the classification standard, the evaluation of a single indicator can be expressed as a five-level fuzzy subset $R_j = \{0, \cdots, x, 1-x, \cdots, 0\}$. $x$ is the $i$th component ($i = 1, 2, 3, 4$), which represents the membership degree of the indicator to the $i$th level. $1-x$ is the $i+1$th component, which represents the membership degree of the indicator to the $i+1$th level. Suppose the original value of the $j$th indicator is $Z_j$, $Z_j$ is exactly between the $i$th standard critical value $\mu_{j,i}$ and the $i+1$th standard critical value $\mu_{j,i+1}$. $Z_j$ can be expressed as follows:

$$\mu_{j,i} \times x + \mu_{j,i+1} \times (1-x) = Z_j \tag{2}$$

Taking $R_j$ as the $j$th row vector, then, the fuzzy membership matrixes $R$ of administrative regions can be constructed.

- Step 2. Construct a pairwise judgment matrix of the criterion layer.

A pairwise judgment matrix is constructed, by means of fuzzy coincident matrix. Suppose the upper target is $V$, and the lower target after decomposing it is $V_1, V_2, \cdots, V_n$. For $V$, the pairwise judgment matrix of the lower sub-target is $A$, and the element $a_{i,j}$ in the $i(i = 1, 2, 3)$th row and $j(j = 1, 2, 3)$th column represents the relative importance of the factor $V_i$ to the factor $V_j$, in terms of the upper target $V$. A pairwise judgment matrix of the evaluation system is described as follows:

If $a_{i,j} > 1$, it means that the factor $V_i$ is more important than the factor $V_j$; otherwise, the factor $V_i$ is less important than the factor $V_j$. If $a_{i,j} = 1$, it means that the factor $V_i$ and the factor $V_j$ are equally important.

Accordingly, this pairwise judgment matrix $A$ can be expressed as follows:

$$A = \begin{bmatrix} a_{1,1} & a_{1,2} & a_{1,3} \\ a_{2,1} & a_{2,2} & a_{2,3} \\ a_{3,1} & a_{3,2} & a_{3,3} \end{bmatrix} \tag{3}$$

where $a_{i,j} > 0$ and $a_{i,j} \times a_{j,i} = 1$.

- Step 3. Calculate the eigenvector and the eigenvalue of the criterion layer.

The square root method is adopted to calculate the eigenvalue and the eigenvector of the pairwise judgment matrix. Firstly, a new vector $M_i$ is obtained by multiplying the elements of the judgment matrix $A$ by rows:

$$M_i = \prod_{j=1}^{n} a_{ij}, i = 1, 2, \cdots, n \tag{4}$$

Then, calculate the $n$th root of $M_i$:

$$\overline{W}_i = \sqrt[n]{M_i} \tag{5}$$

Subsequently, standardize $\overline{W}_i$ to obtain the eigenvector:

$$W_i = \frac{\overline{W}_i}{\sum\limits_{j=1}^{n} \overline{W}_j} \tag{6}$$

Finally, calculate the maximum eigenvalue $\lambda_{\max}$:

$$\lambda_{\max} \approx \sum_{i=1}^{n} \frac{(AW)_i}{nW_i} \tag{7}$$

where $AW_i$ represents the $i$th component of the vector $AW$.

- Step 4. Single hierarchical arrangement and consistency check of the criterion layer.

The consistency index ($CI$) of the judgment matrix can be expressed as follows:

$$CI = \frac{\lambda_{\max} - n}{n - 1} \tag{8}$$

If $CI = 0$, it means that two factors have complete consistency. If $CI$ is close to 0, it means that two factors have satisfactory consistency. In addition, the greater the $CI$, the more serious the inconsistency between two factors.

The mean random consistency index ($RI$) of the judgment matrix corresponding to the order number $n$, can be obtained through the table below.

Then, the random coincidence coefficient ($CR$) of the judgment matrix can be expressed as follows:

$$CR = \frac{CI}{RI} \tag{9}$$

If $CR < 0.1$, it means that the judgment matrix have good consistency, otherwise the element values of the judgment matrix should be adjusted.

- Step 5. Calculate the weight coefficient of the index layer.

In the same way, repeat calculation procedures mentioned above from Step 2 to Step 4 to calculate the weight coefficients of the index layer, so as to obtain the final weight values of evaluation indicators.

- Step 6. Put forward comprehensive evaluation result.

A comprehensive evaluation equation can be expressed as follows:

$$F = A \times R \times C^T \tag{10}$$

where $F$ represents the comprehensive evaluation result, $A$ represents the weight vector, $R$ represents the fuzzy membership matrix and $C$ represents the evaluation grade vector. $F < 2$, $2 \le F < 3$, $3 \le F < 4$, $4 \le F < 5$ and $5 \le F$ represent the administrative region with very low, low, medium, high and very high water use efficiency, respectively.

## 3. Results and Discussions

### 3.1. Pairwise Judgment Matrix Result

Based on Table 3, applying expert scoring method to measure the importance of different criteria and constructing a judgment matrix of the criterion layer. Then, the pairwise judgment matrix is expressed as follows:

$$A = \begin{bmatrix} 1 & 6 & 3 \\ 1/6 & 1 & 1/2 \\ 1/3 & 2 & 1 \end{bmatrix} \tag{11}$$

**Table 3.** A pairwise judgment matrix of the evaluation system.

| A | B1 | B2 | B3 |
|---|---|---|---|
| B1 (Water Use Situation) | $a_{1,1}$ | $a_{1,2}$ | $a_{1,3}$ |
| B2 (Engineering Measure) | $a_{2,1}$ | $a_{2,2}$ | $a_{2,3}$ |
| B3 (Planting Structure) | $a_{3,1}$ | $a_{3,2}$ | $a_{3,3}$ |

It is obvious to find that the water use situation has the most importance, while the engineering measure has the least importance in the criterion layer. Then, the maximum eigenvalue, the consistency index, the mean random consistency index and the random coincidence coefficient can be calculated according to Equations (4)–(9) and Table 4:

$$\lambda_{\max} = 3, \ CI = 0, \ RI = 0.58, \ CR = 0 \tag{12}$$

**Table 4.** Values of mean random consistency index (*RI*).

| Order Number $n$ | 1 | 2 | 3 | 4 | 5 | 6 | 7 | 8 | 9 |
|---|---|---|---|---|---|---|---|---|---|
| *RI* | 0 | 0 | 0.58 | 0.90 | 1.12 | 1.24 | 1.32 | 1.41 | 1.45 |

As a result, the judgment matrix of the criterion layer has complete consistency. Similarly, pairwise judgment matrixes of the index layer and their consistency check are shown in Table 5 (i.e., evaluation indicators belong to the criterion layer of B1) and Table 6 (i.e., evaluation indicators belong to the criterion layer of B3). The judgment matrixes of the index layer have complete consistency, from the perspective of B1 (water use situation) and B3 (planting structure), respectively.

**Table 5.** Pairwise judgment matrixes of the index layer and their consistency check (evaluation indicators belong to the criterion layer of B1).

| B1 | C1 | C2 | C3 | C4 | C5 | $M_i$ | $\overline{W}_i$ | $W_i$ | AW | $\lambda_{max}$ | CR |
|----|----|----|----|----|----|-------|------------------|-------|-----|-----------------|-----|
| C1 | 1 | 1 | 3 | 2 | 3 | 18 | 1.78 | 0.32 | 1.57 | | |
| C2 | 1 | 1 | 3 | 2 | 3 | 18 | 1.78 | 0.32 | 1.57 | | |
| C3 | 1/3 | 1/3 | 1 | 2/3 | 1 | 0.07 | 0.59 | 0.10 | 0.5 | 5 | 0 |
| C4 | 1/2 | 1/2 | 2/3 | 1 | 2/3 | 0.56 | 0.89 | 0.16 | 4/5 | | |
| C5 | 1/3 | 1/3 | 1 | 2/3 | 1 | 0.07 | 0.59 | 0.10 | 1/2 | | |

**Table 6.** Pairwise judgment matrixes of the index layer and their consistency check (evaluation indicators belong to the criterion layer of B3).

| B3 | C7 | C8 | $M_i$ | $\overline{W}_i$ | $W_i$ | AW | $\lambda_{max}$ | CR |
|----|----|----|-------|------------------|-------|-----|-----------------|-----|
| C7 | 1 | 2 | 2 | 1.41 | 0.67 | 1.33 | 2 | 0 |
| C8 | 1/2 | 1 | 1/2 | 0.71 | 0.33 | 0.67 | | |

Accordingly, the weight values of evaluation indicators in the index layer are listed in Table 7. Indicators C1 (the utilization coefficient of irrigation water) and C2 (weighting irrigation quota) have the highest weight value, both of which are 0.21. Indicators C3 (grain production per cubic meter of irrigation water), C5 (water-saving irrigation rate) and C8 (water-consumption proportion of crops) have the lowest weight value, all of which are 0.07. In other words, indicators C1 and C2 are more important than C4, as well as C6. Compared to other indicators, indicators C3, C5 and C8 have the least importance.

**Table 7.** The weight values of evaluation indicators in the index layer.

| Evaluation Indicator | B1 (0.67) | B2 (0.11) | B3 (0.22) | Weight Value |
|----------------------|-----------|-----------|-----------|--------------|
| C1 | 0.32 | 0 | 0 | 0.21 |
| C2 | 0.32 | 0 | 0 | 0.21 |
| C3 | 0.1 | 0 | 0 | 0.07 |
| C4 | 0.16 | 0 | 0 | 0.11 |
| C5 | 0.1 | 0 | 0 | 0.07 |
| C6 | 0 | 1 | 0 | 0.11 |
| C7 | 0 | 0 | 0.67 | 0.15 |
| C8 | 0 | 0 | 0.33 | 0.07 |

*3.2. Comprehensive Evaluation Result*

According to Equation (10), agricultural WUE values of 14 administrative regions can be determined, and their corresponding classification standards are displayed in Table 8. Distribution map of agricultural WUE values in administrative regions is shown in Figure 3.

**Table 8.** Agricultural water use efficiency values and evaluation grades of 14 administrative regions.

| Region | Efficiency Value | Evaluation Grade | Region | Efficiency Value | Evaluation Grade |
|--------|------------------|------------------|--------|------------------|------------------|
| Urumqi | 3.25 | Medium | Aksu | 2.36 | Low |
| Karamay | 3.44 | Medium | Kizilsu | 2.70 | Low |
| Turpan | 3.09 | Medium | Kashgar | 1.93 | Very Low |
| Hami | 4.26 | High | Hotan | 1.99 | Very Low |
| Changji | 4.19 | High | Yili | 2.51 | Low |
| Bortala | 3.77 | Medium | Tacheng | 3.14 | Medium |
| Bayingol | 2.96 | Low | Altay | 2.93 | Low |

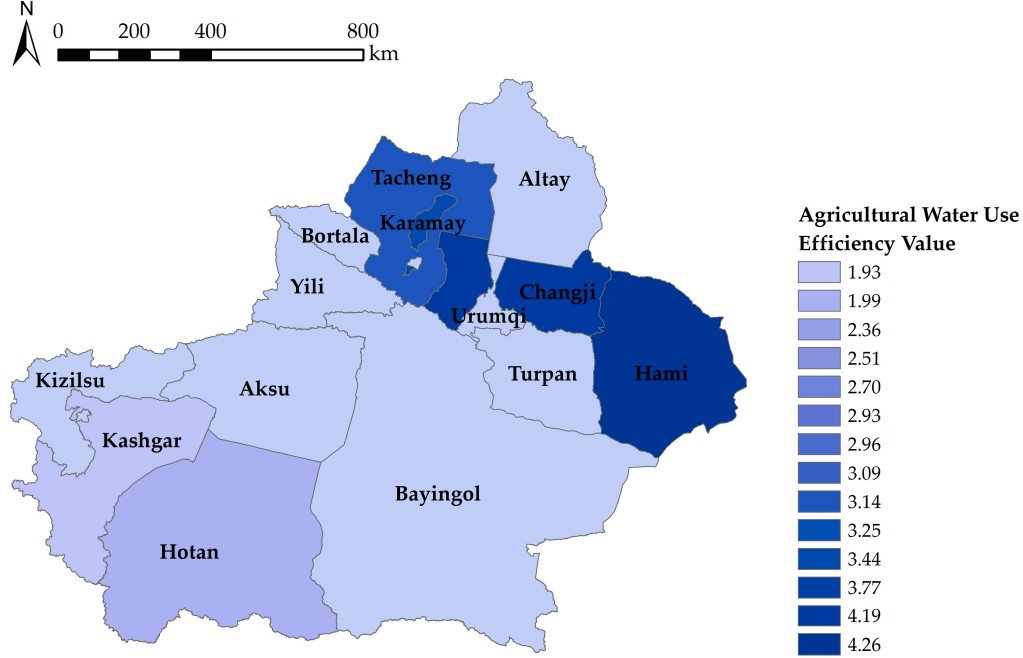

**Figure 3.** Distribution map of agricultural water use efficiency values in 14 administrative regions.

From the classification standard of regions, it is evident that two regions including Hami and Changji, have high agricultural WUE; five regions including Urumqi, Karamay, Turpan, Bortala and Tacheng, have medium agricultural WUE; five regions including Bayingol, Aksu, Kizilsu, Yili and Altay, have low agricultural WUE; while two regions including Kashgar and Hotan, have very low agricultural WUE. From the agricultural WUE of each region, Hami has the highest WUE, with an efficiency value of 4.26. Kashgar has the lowest WUE, with an efficiency value of 1.93. The average WUE value in northern Xinjiang (i.e., Urumqi, Karamay, Changji, Bortala, Yili, Tacheng, Altay) is 3.32, while that in southern Xinjiang (i.e., Bayingol, Aksu, Kizilsu, Kashgar, Hotan) and eastern Xinjiang (i.e., Turpan, Hami) are 2.39 and 3.68, respectively. Thus, agricultural WUE in eastern Xinjiang is greater than that in western Xinjiang, and that in southern Xinjiang is the lowest.

In order to compare the evaluation indicators of different regions, fuzzy membership values between these regions with high WUE (i.e., Hami, Changji) and very low WUE (i.e., Kashgar, Hotan) are selected to analyze their differences. Fuzzy membership values of four regions are shown in Figure 4.

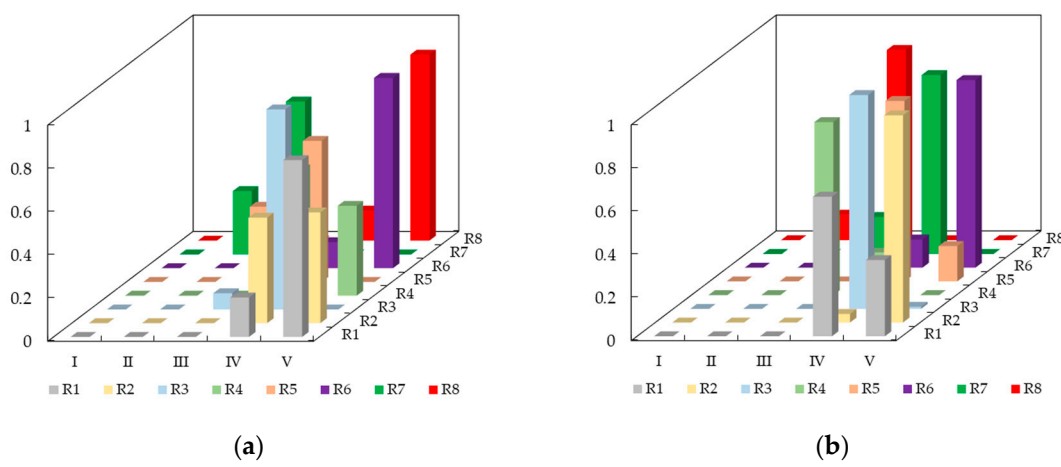

(**a**)                                        (**b**)

**Figure 4.** *Cont.*

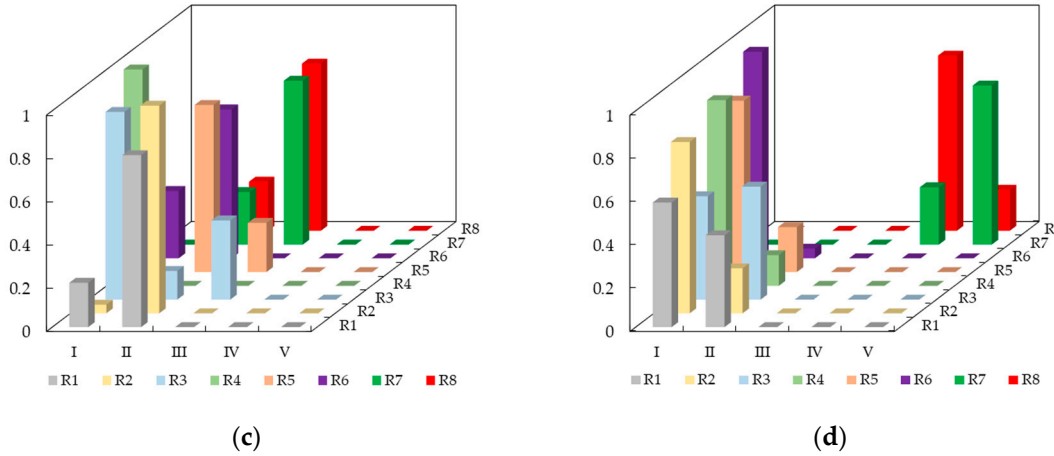

**Figure 4.** Fuzzy membership values of four administrative regions: (**a**) Hami; (**b**) Changji; (**c**) Kashgar; (**d**) Hotan.

As shown in Figure 4, most of indicators in high WUE regions are located between the grade III and V, except the indicators C7 (the planting proportion of crops) in Hami and C8 (the water-consumption proportion of crops) in Changji. On the contrary, most of indicators in very low WUE regions are located between the grade I and III, except the indicators C7 and C8 in Hotan. When it comes to the actual values and the weight of indicators, the weighting irrigation quota (C2) in Hami and Changji are 436 m$^3$/mu and 380 m$^3$/mu, while that in Kashgar and Hotan are 755 m$^3$/mu and 849 m$^3$/mu. Moreover, considering that the indicator C2 has the highest weight value, 0.21, it reveals that the weighting irrigation quota is an important factor affecting agricultural WUE in these four regions. Equally, the grain production per cubic meter of irrigation water (C3), the agricultural water consumption for ten thousand yuan output value (C5) and the water-saving irrigation rate (C6) are also important factors that make differences on WUE in these four regions.

### 3.3. Discussions on Improving Agricultural Water Use Efficiency

Studies demonstrated that there is room for improvement in WUE, particularly for these regions in southwest Xinjiang, such as Kashgar and Hotan. Therefore, Kashgar is taken as a case study to analyze variations of WUE under different indicators scenarios, by means of the factor analysis method. It is assumed that agricultural water consumption will be reduced by 10% on the original basis, with taking the principle of total water consumption control into consideration. In Water Resources Bulletin of Xinjiang Uygur Autonomous Region in 2015, total water consumption in Kashgar is 11.90 billion m$^3$, while agricultural water consumption in this region is 11.63 billion m$^3$, accounting for 97.7% of regional total water consumption. Then, the optimized agricultural water consumption will become 10.63 billion m$^3$, accounting for 87.9% of regional total water consumption.

It is certain that the optimization of agricultural water consumption will bring changes to some indicators. Taking the indicator C2 (the weighting irrigation quota) as an example, the value of the weighting irrigation quota will decrease from 755 to 680 m$^3$/mu. Similarly, the indicator C3 (the grain production per cubic meter of irrigation water) will increase from 0.57 to 0.64 kg/m$^3$; the indicator C4 (the proportion of agricultural water) will decrease from 97.7 to 87.9%; and the indicator C5 (the agricultural water consumption for ten thousand yuan output value) will decrease from 5545 to 4991 m$^3$. Meanwhile, other indicators (i.e., C1, C6, C7, C8) will remain unchanged. Comparisons of evaluation indicators values in Kashgar are listed in Table 9.

**Table 9.** Evaluation indicators values before and after optimization in Kashgar.

| Evaluation Indicator | Weight Value | Indicator Value Before and After Optimization | |
| :---: | :---: | :---: | :---: |
| | | Before Optimization | After Optimization |
| C1 | 0.21 | 0.466 | 0.466 |
| C2 ($m^3$/mu) | 0.21 | 755 | 680 |
| C3 ($kg/m^3$) | 0.07 | 0.57 | 0.64 |
| C4 (%) | 0.11 | 97.7 | 87.9 |
| C5 ($m^3$) | 0.07 | 5545 | 4991 |
| C6 | 0.11 | 0.203 | 0.203 |
| C7 (%) | 0.15 | 36.4 | 36.4 |
| C8 | 0.07 | 0.554 | 0.554 |

The optimized indicator values are substituted into Equations (2)–(10), to calculate the optimized agricultural WUE value in Kashgar. It turns out that agricultural WUE value after optimization is 2.50, while that before optimization is just 1.93. At the same time, Kashgar is upgraded from a region with very low WUE to a region with low WUE.

Except for reducing agricultural water consumption, other measures, such as improving the utilization coefficient of irrigation water (C1) and increasing the water-saving irrigation rate (C6), are equally important to raise agricultural WUE in Kashgar. As far as the indicator C1 concerned, the utilization coefficient of irrigation water in Kashgar is 0.466, which is determined by the utilization coefficient of irrigation water for large irrigation areas (0.465) and medium irrigation areas (0.475). However, in Technical standard for water-saving irrigation project, it is mentioned that the utilization coefficient of irrigation water for large and medium irrigation areas should not be lower than 0.55 and 0.65, respectively, which indicate that there exists tap between water utilization situation in Kashgar and the national standard. When it comes to the indicator C6, the water-saving irrigation rate is 0.203 in Kashgar, while that in Karamay, Bortala and Altay are 0.741, 0.798 and 0.784, respectively. Furthermore, water-saving irrigation techniques in Kashgar are obviously behind the average level of Xinjiang, because the mean water-saving irrigation rate is 0.461 in 14 administrative regions. As a consequence, it is essential to implement water-saving irrigation projects and promote water-saving irrigation technologies in this region, which may make a big difference to agricultural WUE in Kashgar.

## 4. Conclusions

In view of quantifying agricultural water use efficiency (WUE) of 14 administrative regions in Xinjiang, a coupled model composed of the analytic hierarchy process (AHP) method and the fuzzy evaluation method is applied to evaluate the situation of agricultural water utilization. The main conclusions of this study are as follows: (1) Indicators C1 (the utilization coefficient of irrigation water) and C2 (the weighting irrigation quota) have the most importance, while indicators C3 (the grain production per cubic meter of irrigation water), C5 (the agricultural water consumption for ten thousand yuan output value) and C8 (the water-consumption proportion of crops) have the least importance. The former weight values are all 0.21, while the latter weight values are all 0.07. (2) In terms of administrative regions, Hami has the highest agricultural WUE, while Kashgar has the lowest agricultural WUE. Agricultural WUE in eastern Xinjiang is greater than that in western Xinjiang, and that in southern Xinjiang is the lowest. (3) Most of indicators in high WUE regions are located between the grade III to V, except the indicators C7 in Hami and C8 in Changji. Most of evaluation indicators in low WUE regions are located between the grade I to III, except the indicators C7 and C8 in Hotan. Indicators C2, C3, C5 and C6 are important factors that make differences to these four regions with high or low WUE. (4) Kashgar is selected as a case study to analyze its variations of WUE under different indicators scenarios. Supposing agricultural water consumption will be reduced by 10%, agricultural WUE value will raise from 1.93 to 2.50, and Kashgar is upgraded from a region with very low WUE to a region with low WUE.

The evaluation results of this study are beneficial for providing support for reforming agricultural water use and promoting sustainable agricultural development in Xinjiang. Nevertheless, due to the limited data collected, the evaluation system constructed and the evaluation indicators selected are not synthetical enough in this study, which may influence practical applications when this coupled model is adopted in other similar areas. Furthermore, proposing a more reasonable method for determining the weight of evaluation indicators will still be the focus of future research.

**Author Contributions:** P.S., X.W. and C.W. conducted the research and designed and wrote the paper; M.L., L.C. and L.K. helped to revise the paper; X.L. and H.W. gave the comments. All authors have read and agreed to the published version of the manuscript.

**Funding:** This research was funded by the National Natural Science Foundation of China (51709275), the Young Elite Scientists Sponsorship Program by CAST (2019QNRC001), the Fundamental Research Funds of IWHR (WR0145B012020), and the National Key Research and Development Plan of China (2018YFC0407405).

**Acknowledgments:** The anonymous reviewers and the editor are thanked for providing insightful and detailed reviews that greatly improved the manuscript.

**Conflicts of Interest:** The authors declare no conflict of interest.

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
