# Peer review of "Analysis of Agricultural Water Use Efficiency Based on Analytic Hierarchy Process and Fuzzy Comprehensive Evaluation in Xinjiang, China"

_water, doi:10.3390/w12113266_

Round 1

Reviewer 1 Report

Authors have investigated a very interesting topic. The presentation is great however, the introduction should be improved with some practical data from similar studies. Also, authors should present some agricultural water use efficiency values across the study area related to the grown crop across the different regions of the study area.

Also the discussion section seems not as a discussion. The section must be improved with similar study results from different parts of the globe and some real field data from the study area across one or two decades to show if there is already increasing or decreasing trend in water use efficiency. Specific comment are marked on the attached pdf file.

Author Response

Thank you for your anonymous comments concerning our manuscript entitled “Analysis of Agricultural Water Use Efficiency Based on Analytic Hierarchy Process and Fuzzy Comprehensive Evaluation in Xinjiang, China” (No. water-966360). Those comments are all valuable and very helpful for revising and improving our paper, as well as the important guiding significance to our researches. We have studied the comments carefully and made revisions accordingly, which we hope meet with your approval.

Revised portion are marked in yellow in the paper and the detailed corrections are listed below point by point (in the attachment).

Reviewer 2 Report

Keywords: Please consider delete: analytic hierarchy process; fuzzy comprehensive; Xinjiang, that are already in the title. And also “water management” that could be included

agricultural water use efficiency (WUE)

INTRODUCTION

lines 45-47: what happens in the last 15 years?

lines 49-51 " ...... and uneven spatial patterns of precipitation changes may enhance drought impacts in the 21st century [10, 11]." there are a large current literature that mention this fact for granted. I suggest that you could cited some IPCC reports

lines 60-72: commonly known, maybe you could shorten this part

lines 99-105: difficult reading lines

MATERIALS AND METHODS

This section is very complete and well structured, but it is too long. I would suggest to revise it and make it shorter.

water use efficiency: please write (WUE)

155-159: the population is use for material in this paper? these data are for introduction section

line 176: WUE??

line 212: what are the advanced water-saving indicators at home and abroad?

Table 1: I don't understand how do you calculate the values of table 1

Lines 238-247: perhaps this is discussion of the methods and not methodology

RESULTS AND DISCUSSIONS

Lines 366-371 difficult reading and understanding lines

lines 376-381: is a conclusion??

Author Response

(The authors gave the same response as above.)

Round 2

Reviewer 1 Report

Authors have improved the manuscript and need just one last editing to get ride of any misspelling across the manuscript. There a very minor correction to be done.

Author Response

Minor revised portions are marked in red in the paper, which can be obtained in detail in the revised manuscript. We are grateful for anonymous reviewers to provide insightful and detailed reviews that greatly improved the manuscript.

Reviewer 2 Report

thank for considering my suggestions in your paper. congratulation for your paper

Author Response

(The authors gave the same response as above.)
